# High-Speed Centrifugal Spinning Polymer Slip Mechanism and PEO/PVA Composite Fiber Preparation

**DOI:** 10.3390/nano13071277

**Published:** 2023-04-04

**Authors:** Peiyan Ye, Qinghua Guo, Zhiming Zhang, Qiao Xu

**Affiliations:** 1School of Mechanical Engineering and Automation, Wuhan Textile University, Wuhan 430200, China; 2Hubei Digital Textile Equipment Key Laboratory, Wuhan Textile University, Wuhan 430200, China

**Keywords:** centrifugal spinning, slippage, composite nanofibers, fiber morphology

## Abstract

Composite nanofibers with excellent physical and chemical properties are widely used in new energy, biomedical, environmental, electronic, and other fields. Their preparation methods have been investigated extensively by many experts. High-speed centrifugal spinning is a novel method used to fabricate composite nanofibers. The slip mechanism of polymer solution flows is an important factor affecting the morphology and quality of composite nanofibers prepared by high-speed centrifugal spinning. As the polymer solution flows, the liquid wall slip occurs inside the nozzle, followed by liquid–liquid interface slip and gas–liquid interface slip. The factors affecting polymer slip were investigated by developing a mathematical model in the nozzle. This suggests that the magnitude of the velocity is an important factor that affects polymer slip and determines fiber quality and morphology. Under the same rotational speed, the smaller the nozzle diameter, the greater the concentration of velocity distribution and the smaller the diameter of the produced composite nanofibers. Finally, PEO/PVA composite nanofibers were prepared using high-speed centrifugal spinning equipment at 900–5000 rpm and nozzle diameters of 0.2 mm, 0.4 mm, 0.6 mm, and 0.8 mm. The morphology and quality of the collected PEO/PVA composite nanofibers were analyzed using scanning electron microscopy (SEM) and TG experiments. Then, the optimal parameters for the preparation of PEO/PVA composite nanofibers by high-speed centrifugal spinning were obtained by combining the external environmental factors in the preparation process. Theoretical evaluation and experimental data were provided for the centrifugal composite spinning slip mechanism and for the preparation of composite nanofibers.

## 1. Introduction

Nanofibers exhibit excellent physical and chemical properties and are widely considered to be nanomaterials with great application potential [1]. The most distinguishing features of nanofibers are their superior surface area ratio and interconnected porous structure. These properties also ensure that nanofibers exhibit many desirable properties for advanced application [2]. Currently, nanofibers can be prepared from various materials, such as natural polymers [3], synthetic polymers [4], semiconductor nanoscale materials [5], and composite nanomaterials [6]. The composite nanofiber has received a lot of attention because of its outstanding performance. In recent years, composite nanofibers have been widely used in various fields such as agriculture, tissue engineering, biomedical engineering, pressure sensors, energy, electronics, environment, and optics [7,8,9,10,11,12,13,14]. With the rapidly evolving nature of these fields, the production of composite nanofibers with controlled shape, size, and surface properties becomes important for different practical applications [15]. Therefore, the critical technology for the composite nanofiber preparation method and the development technology of equipment have become the focus of research. Traditional methods for nanofiber preparation mainly include electrostatic spinning, melt spinning, wet spinning, and centrifugal spinning [16,17,18,19,20]. Among these preparation methods, electrostatic spinning is a relatively stable and simple technique that can be used to prepare fibers with very high surface-area-to-volume ratio, adjustable porosity, a ductility that conforms to various sizes and shapes, and the ability to control nanofiber composition. Accordingly, electrostatic spinning has become the most mature and widely adopted technology [21]. However, low preparation efficiency, complex processing, and high-voltage electric field limit its potential industrial applications. Novel methods of nanofiber fabrication for industrial applications need to be explored [22]. In contrast to electrostatic spinning, high-speed centrifugal spinning technology is a promising method for preparing nanofibers with high productivity [23].

Compared with several traditional methods of preparation methods, high-speed centrifugal spinning is simple and easy to use to produce nanofibers from various materials efficiently and cost-effectively [24]. A series of structural optimizations of the centrifugal spinning device were carried out by changing the internal structure of the tank and selecting different nozzle shapes for the efficient preparation of composite nanofibers with significantly improved performance using the high-speed centrifugal spinning method, to meet diverse needs of various fields as the focus of research.

A schematic illustration of the planar structure of the high-speed centrifugal spinning device is presented in Figure 1. It consists of a drive motor, motor speed controller, nozzle, container, collection tray, and a series of rod collection devices. During centrifugal spinning, the polymer solution is injected into the rotating tank. When the applied centrifugal velocity exceeds the surface tension of the solution, a jet is formed and ejected from the nozzle; the jet elongates and the solution evaporates until it is stretched and deposited on the collection bar [25,26]. It is possible to control the porosity and diameter of polymer fibers by varying the nozzle geometry, rotational speed, and polymer solution. In addition, the high rotational speed facilitates fast and scalable fiber fabrication, significantly increasing the productivity [27].

In the early stages of hydrodynamic development, a few simple experimental observations were consistent with the assumption of no slip, so the slip phenomenon received little further attention. In recent years, with the rapid development of nanofabrication technologies, slip has played an important role in many industrial fields, such as electrochemical devices, polymer processing, and microfluidics [28,29]. The role of slip is also reflected in the study of molecular dynamics [30]. This requires a more in-depth understanding of slip as a physical phenomenon.

The slip problem was first introduced by Mooney [31], who experimented with capillaries of different radii to obtain flow curves. It was found experimentally that once the stress exceeds a critical value, the flow profile depends on the capillary radius, which indicates the occurrence of wall slip. This technique is also known as the Mooney technique and is still widely used in various studies to measure slip velocity. Craig [32] described the extent of slip in terms of the magnitude of slip length. Its function was attributed to the surface approach velocity and fluid viscosity, which has implications for closed biological systems, the permeability of microporous media, and the lubrication of nanomachines, as well as in the micro-control of fluid flows.

Hatzikiriakos [33] investigated different types of complex fluids via different slip mechanisms, including the adhesive slip of polymers on walls; the apparent slip due to the formation of polymer solutions, solid suspensions/fiber suspensions, dispersions, and several paste materials; the lubrication slip of gels, microgels, and colloidal suspensions. Thus, a systematic study of several types of complex fluid slip is needed to determine its effect on particle size, concentration, and wall-wetting properties, and the effects of the slip in such fluids with appropriate models.

You et al. [34] analyzed the effect of boundary slip on the flow stability of finite miscible/immiscible liquid–liquid laminar micro-channels. In this approach, the boundary slip was considered based on the Navier slip assumption and the finite-miscible liquid–liquid interface was modeled using the double-film model. The boundary slip improved the stability of the finely layered micro-channel flow and the stability of this flow was controlled by selecting the appropriate boundary slip, contact fluid, and interface location.

Recent studies have shown that spinning parameters such as motor speed, nozzle length, nozzle diameter, nozzle structure, and solution concentration directly affect the morphological distribution of nanofibers during high-speed centrifugal spinning [35]. Our study investigated the slip of polymer solutions during centrifugal compound spinning. The various relationships between spinning fluid parameters, equipment parameters, environmental parameters, and slip were analyzed experimentally. Further, the influence of the polymer slip mechanism on the preparation of composite fibers was verified.

## 2. Slip Modeling of Polymers in Centrifugal Composite Spinning

In previous studies, the no-slip boundary condition was one of the core concepts of fluid mechanics. This theory is reflected in a vast number of experimental results and utilized in many studies, which led to the neglect of the slip theory in related studies. With the significant improvement in experimental techniques, the slip phenomenon has been clearly demonstrated. The no-slip boundary condition is usually not maintained on the microscopic scale at sufficiently high shear rates. Many complex fluids exhibit inconsistencies with the no-slip boundary conditions, as most spinning solutions are non-Newtonian fluids characterized by a certain degree of viscoelasticity. Therefore, it is important to accurately determine the fluid properties of the polymer. In the case of high-speed rotational motion, the viscosity of the polymer solution of the wall-adherent layer varies with the shear rate, so both polymer solutions slip with the nozzle wall, the solution contact surface, and the air. The liquid–wall slip of polymer solutions during centrifugal composite spinning improves the extrusion and expansion of composite nanofibers and thins the average diameter of composite nanofibers. The liquid–liquid slip changes the internal structure of the composite nanofibers, resulting in composite nanofibers with different properties. Therefore, the polymer slip mechanism has a remarkably important effect on both polymer flow and deformation.

The “slip length” is the most common concept used currently to quantify non-liquid slip on solid surfaces. This theoretical model was introduced into the nozzle to describe the slip on liquid and solid surfaces using the slip length:(1)$$u_{s}=b\frac{\partial v}{\partial z}{|}_{wall}$$
where *b* is the slip length (the distance outside the liquid/solid interface, when the velocity of the liquid can be extrapolated to zero). If *b* = 0, then there is no-slip boundary condition. However, if *b* ≠ 0, there is a slip boundary condition. *u_s_* indicates the liquid wall slip velocity; ∂v/∂z represents the shear rate of the polymer solution at the nozzle wall.

A parametric model was built in the nozzle, as shown in Figure 2. The motion of the solution in the nozzle was studied and analyzed. As the flow in the nozzle class is axisymmetric, the analysis of the motion was not considered. Accordingly, based on the motion of the columnar coordinates to simplify the finishing, the solution in the nozzle wall shear stress can be obtained as follows:(2)$$\tau_{w}=\frac{\Delta pd}{2\Delta L}$$

In Equation (2) above, r is the length of the distance between the nozzle wall and the middle partition; Δp indicates the fluid pressure drop; ΔL denotes the length of the nozzle. The strain rate of the polymer solution is determined in terms of the strain rate of a power-law fluid:(3)$$f(\tau )={\left(\frac{k}{\tau }\right)}^{-\frac{1}{n}}$$
where *k* and *n* are the rheological parameters. As the polymer solution is a non-Newtonian fluid, it does not conform to Newton’s law of internal friction. Considering it as a power-law fluid, its flow velocity distribution is determined:(4)$$u=\frac{r}{\tau_{w}}\int_{\tau }^{\tau_{w}}f(\tau )d\tau$$

The shear stress also increases as the nozzle radius increases. The polymer flow rate distribution starts to decrease. Substituting Equation (3) into Equations (4) and (5), the following can be obtained:(5)$$u=\frac{n}{n+1}{\left(\frac{\Delta p}{2k\Delta L}\right)}^{\frac{1}{n}}(r^{\frac{n+1}{n}}-d^{\frac{n+1}{n}})$$

The flow rate *Q* in the nozzle is:(6)$$Q=\frac{\pi r^{3}}{\tau_{w}{}^{3}}{\int_{0}^{\tau_{w}}\left(\frac{k}{\tau }\right)}^{-\frac{1}{n}}\tau^{2}d\tau$$

Substituting Equations (2) and (3) into Equation (6) gives:(7)$$Q=\frac{n\pi }{3n+1}{\left(\frac{\Delta p}{2k\Delta L}\right)}^{\frac{1}{n}}r^{\frac{3n+1}{n}}$$

The size of the nozzle diameter affects the velocity distribution and flow rate of the polymer solution in the nozzle as well as polymer jet formation. It also indicates differences in the quality of the composite nanofibers obtained via nozzles of different diameter.

High-speed centrifugal spinning technology uses the centrifugal force generated by high-speed rotation as the driving force to produce nanofibers, primarily via high-speed rotation of the polymer in solution as the spinning material. At this time, the nozzle is in a high-speed rotational motion. The nozzle compound spinning solution in the multi-field exhibits a coupling effect along the nozzle outward movement. As the two different polymer solutions are continuous media, they permeate and couple with each other. However, the different kinematic properties lead to sliding of the two polymer solutions against each other, creating a two-phase flow. The flow of spinning solutions with different characteristics and concentrations generates a contact surface between the two solutions. The solution molecules continuously diffuse through the contact surface, forming a mixed layer with limited miscibility between the two phases, as shown in Figure 3. After the solution enters the nozzle, two different solution layers and a mixed layer appear inside the nozzle. Because of their different properties, the movement of each phase also differs. The nozzle is generated within the liquid–liquid slip. The slip between polymer I and polymer II in the mixed layer was selected as the study goal, as shown in Figure 4.

The continuity equation of the *q* phase is determined by:(8)$$\frac{\alpha }{\alpha t}\left(\alpha_{q}\rho_{q}\right)+\nabla \cdot \left(\alpha_{q}\rho_{q}\vec{v_{q}}\right)=\sum_{p=1}^{n}\overset{\cdot }{m_{pq}}$$
where $\alpha_{q}$ is the volume fraction of the *p* phase; $\rho_{q}$ is the physical density of the *p* phase; ${\vec{v}}_{q}$ represents the velocity of the *p* phase; ${\vec{m}}_{pq}$ denotes the mass transfer from phase *p* to phase *q*. Using the mass equation, Equation (9) can be derived:(9)$$\sum_{p=1}^{n}\overset{\cdot }{m_{pq}}=0$$

The momentum conservation equation is as follows:(10)$$\begin{array}{l} \frac{\alpha }{\alpha t}\left(\alpha_{q}\rho_{q}\vec{v_{q}}\right)+\nabla \cdot \left(\alpha_{q}\rho_{q}\vec{v_{q}}\vec{v_{q}}\right)=-\alpha_{q}\nabla p+\\ \nabla \cdot \bar{\bar{\tau_{q}}}+\sum_{p=1}^{n}\overset{\cdot }{m_{pq}}\vec{v_{pq}}+\alpha_{q}\rho_{q}\vec{F_{q}} \end{array}$$

In Equation (10) above, $\bar{\bar{\tau_{q}}}$ is the pressure strain tensor, ${\vec{F}}_{q}$ denotes the external volume force, and ${\vec{v}}_{pq}$ represents interphase velocity.

In the initial stage of jet injection, the power wave is generated by fluctuations between the liquid–liquid two-phase interface in the mixed layer of the polymer solutions due to flow instability, as the polymer solution has yet to be solidified. The liquid–liquid interface with fiber length dx was selected for analysis, as shown in Figure 5. The two-phase fluid inside the nozzle flows at $u_{1}$ and $u_{2}$ velocities, assuming that there is no volumetric force in the direction of flow, but under the influence of the solution concentration, the change in static pressure through the nozzle tube wall generates a force f in the direction of flow. The static force per unit width acting on the micro-element body of solution 1 is:(11)$$F_{1}=\left(1-\alpha \right)H\delta p$$

The static force per unit width acting on the micro-element body of solution 2 is:(12)$$F_{2}=\alpha H\left(\delta p-g\left(\rho_{2}-\rho_{1}\right)H\delta \alpha \right)$$

The total force per unit volume of solution 1 minus the total force per unit volume of fluid 2 can be expressed as:(13)$$f_{1}-f_{2}=\left(\rho_{2}-\rho_{1}\right)gH\frac{\partial \alpha }{\partial z}$$
and the speed of the power wave is:(14)$$U={\left[\frac{-{\left(u_{1}-u_{2}\right)}^{2}}{\frac{\alpha }{\rho_{1}}+\frac{1-\alpha }{\rho_{2}}}+\left(\rho_{2}-\rho_{1}\right)gH\right]}^{\frac{1}{2}}{\left(\frac{\rho_{1}}{1-\alpha }+\frac{\rho_{2}}{\alpha }\right)}^{-\frac{1}{2}}$$

The flow tends to stabilize when the slip velocity of the two solutions has the following relationship:(15)$$\left(u_{1}-u_{2}\right)={\langle \left[\left(\rho_{2}-\rho_{1}\right)gH\right]\left(\frac{\alpha }{\rho_{2}}+\frac{1-\alpha }{\rho_{1}}\right)\rangle }^{\frac{1}{2}}$$

Therefore, the slip velocity of the liquid–liquid two-phase is an important factor determining the fluctuation across the liquid–liquid two-phase interface, and the fluctuation occurring at the two-phase interface will lead to the instability of the flow of the two polymer solutions, which in turn will affect the preparation of the composite nanofibers and the quality of the fibers after forming.

Centrifugal forces are generated when the nozzle is in high-speed rotational motion, which causes the polymer solution to be subjected to a high-speed airflow from the time it forms a jet until after it leaves the nozzle. When the polymer solution forms a jet leaving the nozzle, it is stretched and evaporated to form composite nanofibers that are eventually collected. A gas–liquid layer is generated between the jet formed by the polymer and the air, resulting in gas–liquid slip. The micro-element dx at the fiber–gas–liquid interface in the nozzle was selected for analysis. Gas–liquid slip has a phase interface, with interactions at the interface as well as between the gas–liquid phases. Energy transfer also occurs across the gas–liquid phase, but both the gas-phase medium and the liquid-phase medium have their own separate physical characteristics, so the basic equations of each phase should be established to investigate the gas–liquid slip. The gas–liquid interface with fiber length dx was selected for analysis, as shown in Figure 6.

Because the polymer solution undergoes stretching and evaporation, the conversion of liquid to gas under thermodynamic equilibrium is expressed as:(16)$$G_{e}=q\frac{P_{p}}{\gamma }$$
where *q* is the heat flow entering from the wall; γ is the latent heat of vaporization; $P_{p}$ is the gas–liquid interface perimeter. Thus, the mass conservation equation for the liquid phase is:(17)$$\frac{\partial }{\partial t}\left[\rho_{L}\left(1-\alpha \right)A\right]+\frac{\partial }{\partial z}\left[\rho_{L}u_{L}\left(1-\alpha \right)A\right]=-G_{e}$$

In Equation (17) above, *A* is the flow cross-sectional area and α denotes the cross-sectional air content rate. Similarly, the mass conservation equation for the vapor phase can be expressed as:(18)$$\frac{\partial }{\partial t}\left(\rho_{G}\alpha A\right)+\frac{\partial }{\partial z}\left(\rho_{G}u_{G}\alpha A\right)=G_{e}$$

Also from the mass flow rate calculation equation:(19)$$G=\rho_{G}u_{G}\alpha +\rho_{L}u_{L}\left(1-\alpha \right)$$

The continuity equation for the gas–liquid two-phase mixture at the time of gas–liquid slip is obtained by combining Equations (17)–(19):(20)$$\frac{\partial }{\partial t}\left(\rho_{LG}A\right)+\frac{\partial }{\partial z}\left(GA\right)=0$$
where $\rho_{LG}$ is the density of the gas and liquid phases. The velocity of gas–liquid slip is then obtained from the mass flow rate equation and the mass gas content *x*:(21)$$u_{G}-u_{L}=\frac{G\left[x\left(1-\alpha \right)\rho_{L}-\left(1-x\right)\alpha \rho_{G}\right]}{\alpha \left(1-\alpha \right)\rho_{L}\rho_{G}}$$

So, the slip model for polymers is summarized as follows: the liquid–wall slip of polymers is related to the size of the nozzle diameter; the liquid–liquid slip velocity affects the stability of the polymer flow; the gas–liquid slip velocity influences the motion of the polymer.

## 3. PEO/PVA Composite Nanofiber Preparation Experiment

### 3.1. PEO/PVA Composite Nanofiber Preparation via Centrifugal Spinning

#### 3.1.1. Spinning Solution Preparation

There is a wide range of materials that can be applied to the preparation of composite nanofibers by high-speed centrifugal spinning, and polyethylene oxide (PEO) and polyvinyl alcohol (PVA) were selected as the centrifugal spinning composite fibers in this experiment. The PEO exhibits excellent properties such as low solution rheology, the combined action with organic solvents, low ash content, and thermo-plasticity. It can be used as a detergent, flocculant, thickener, lubricant, dispersant, aqueous phase reducer, textile slurry, and water-soluble film. PVA is a non-toxic, tasteless, harmless water-soluble polymer, characterized by good density, high crystallinity, and strong adhesion. It is made of flexible and smooth film. It exhibits resistance to oil and abrasion, good gas permeability, as well as water resistance. It has a wide range of uses. It is a natural non-polluting resource. However, at low concentrations, the concentration of the prepared PVA solution viscosity is not adequate to meet the requirements of centrifugal spinning out of the fiber. It exhibits worse performance. However, when the concentration of the prepared PVA solution is large, the solution viscosity is increased. Due to its strong hydrophilicity, the solvent is not easy to volatilize, seriously affecting the structure and performance of centrifugal spinning PVA fiber. The prepared PEO solution exhibits large viscosity at small concentrations. Together with PVA, it can effectively improve the material spinnability for the preparation of composite nanofibers by centrifugal spinning.

In this experiment, specific amounts of PEO and PVA powder were added to water as solutes. PVA was fully dissolved at 90 °C to 100 °C with a magnetic stirrer at 200 rpm for about 5 h. PEO was fully dissolved at room temperature with a magnetic stirrer at 260 rpm for about 10 h. PVA solution with a mass fraction of 9% and PEO solution with a mass fraction of 4% were prepared.

#### 3.1.2. Experimental Procedure for the Preparation of PEO/PVA Composite Nanofibers

The high-speed centrifugal spinning equipment used in the experiments of this paper is shown in Figure 7. The two prepared spinning solutions were added to the jar in which they were stored using a syringe, and then the high-speed centrifugal spinning device was connected to a motor and energized to achieve high-speed centrifugal spinning by controlling the frequency up to 6000 rpm or more. After adjusting the device, spinning was started. As the frequency of the motor was increased to increase the speed, the polymer solution rapidly flew from the jar to the nozzles at both sides under the action of high-speed rotation, and finally were ejected at the nozzles as a jet. With the evaporation of the solvent, the jet was finally collected at the collection device after effective stretching of the PEO/PVA composite nanofibers.

### 3.2. Influence of Spinning Parameters on the Morphology of PEO/PVA Composite Nanofibers

The centrifugal force is generated by high-speed rotation. The difference in the cooling rate of the spinning solution, and the change in concentration, viscous force, and surface tension during the solidification of the jet affect the steady-state motion of the composite jet, which eventually has an important effect on the final morphology of the composite nanofibers. Therefore, the experimental setup parameters of high-speed centrifugal spinning, the selected polymer solution parameters, and the experimental environment are important factors for the preparation and collection of composite nanofibers. In this paper, comparative experiments were performed at different nozzle diameters, different concentrations, and different motor speeds. Finally, the diameter and morphology of the collected composite nanofibers were characterized using scanning electron microscopy (SEM) and TG experiments.

#### 3.2.1. Effect of Rotational Speed on the Morphology of PEO/PVA Composite Nanofibers

In the tank, 9 wt.% PVA solution and 4 wt.% PEO solution were added. The motor was connected and started after adjusting the device.

Spinning experiments indicated that when the rotational speed was below 900 rpm, significant droplets were ejected and no fibers were collected. Because the rotational speed was too small, the polymer solution was not stretched enough and the jet was less stable when rotating in air. Continuing to increase the speed, the PEO/PVA composite nanofibers were prepared and successfully collected in the range of 1200~2100 rpm. Based on the morphology of PEO/PVA composite nanofibers at 1200 rpm and 1500 rpm by SEM, as shown in Figure 8, the fiber structure contained two components, which demonstrates that the collected fibers were composite fibers. The rotational speed was increased continuously to prepare PEO/PVA composite nanofibers. The morphology of PEO/PVA composite nanofibers under different rotational speed conditions was analyzed by SEM.

When the motor speed was in the range of 1200–3000 rpm, the SEM images at 1500 rpm, 2100 rpm, and 2700 rpm were selected, as shown in Figure 9. The morphology and surface quality of PEO/PVA composite nanofibers were enhanced, and the distribution of fiber diameter was more uniform with the increase in speed. Then, the diameters of the PEO/PVA composite nanofibers collected at different rotational speeds were measured, and a histogram was generated to analyze the increasingly uniform diameter of PEO/PVA composite nanofibers. The fiber diameter decreased with the increase in rotational speed. However, as the speed continued to increase beyond this range, the uniformity of the fiber diameter distribution was less evident. At speeds above 3300 rpm, the fiber morphology deteriorated and began to break. At speeds above 3600 rpm, it no longer collected complete fibers at the collection rod. An optimal speed range during fiber preparation ensures an excellent surface quality and diameter distribution of PEO/PVA composite nanofibers with increasing speed, and the fiber diameter is reduced with increasing speed. Therefore, the rotational speed is one of the important elements influencing the diameter and morphology of PEO/PVA composite nanofibers.

#### 3.2.2. Effect of Nozzle Diameter

The initial diameter of the rotating jet is limited by the nozzle diameter as it leaves the nozzle, which in turn affects the diameter of the fibers. In order to investigate the morphological effect of nozzle diameter, PEO/PVA nanofibers were prepared using nozzles of different diameters. The SEM images of PEO/PVA nanofibers prepared with different nozzle diameters are shown in Figure 9. The diameter of PEO nanofibers decreased with decreasing nozzle diameter. The diameter of PEO/PVA composite nanofibers at different nozzle diameters was measured. The average diameter was calculated, and the relationship between the average diameter of PEO/PVA composite nanofibers and the nozzle diameter was obtained, as shown in Figure 10. As shown in Figure 11, the average diameter of PEO/PVA nanofibers decreased with the decrease in nozzle diameter. Therefore, a smaller nozzle diameter facilitates the preparation of nanofibers.

### 3.3. Thermal Stability of PEO/PVA Composite Nanofibers

The thermal decomposition of PEO/PVA films was analyzed at a temperature increase rate of 10 °C/min and a range of 30–800 °C. As shown in Figure 12, the process of heating to 200 °C was mainly the water state of evaporation. The TG curves of both fiber membranes showed a gentle decrease. The total mass loss did not exceed 3%. Between 250 °C and 440 °C, the TG curves of both membranes plummeted, and the total mass loss was about 90%. When the temperature exceeded 500 °C, the combustion of residual materials occurred, resulting in a total mass loss of the samples beyond 90%.

According to the TG curves, the initial decomposition temperature of PEO/PVA composite film was 38.9 °C at 1500 rpm. The initial decomposition temperature of PEO/PVA composite film was 78.9 °C when the speed was increased to 2100 rpm. The thermal decomposition rate of PEO/PVA composite film fabricated at 2100 rpm was lower at 250~440 °C. Thus, the increase in the rotational speed enhanced the mixing of PEO and PVA and improved the thermal stability of the composite nanofibers of PEO/PVA. In addition, the improved spinning condition led to a gradual increase in the decomposition temperature, which enhanced the thermal stability.

### 3.4. Influence of External Environment on Spinning Experiments

The external environment in centrifugal spinning experiments is also one of the most important factors affecting the final morphology and quality of composite nanofibers. In this regard, temperature and humidity are important factors of the external environment affecting the preparation of composite nanofibers. Temperature affects the solidification rate of the spinning jet, and the ability of the spinning jet to solidify quickly and be sufficiently stretched is the key to the preparation and collection of nanofibers. If the spinning jet is incompletely solidified by the time it reaches the collection device, a large number of droplets will appear on the surface of the collected composite nanofibers and stick together, which will lead to a poor morphology and quality of the prepared composite nanofibers. The effect of humidity on the water-soluble spinning solution is attributed to the solidification of PEO upon exposure to water. Therefore, excessive humidity during spinning can lead to solidification of the solution, resulting in nozzle clogging. Therefore, the temperature and humidity of spinning indirectly affected the experimental success of preparing PEO/PVA composite nanofibers. Appropriate adjustment of the environment temperature and humidity based on the physical properties of the spinning material and the device parameters is necessary to ensure a suitable experimental environment for the preparation of composite nanofibers.

## 4. Conclusions

In this paper, the slip mechanism of the polymer in centrifugal composite spinning is analyzed and three slip models are developed. The slip distribution of the polymer is influenced by the rotational speed. As the rotational speed increases, slip occurs first between the nozzle wall and the polymer solution, then between the interfaces of the two different polymer solutions, and finally at the gas–liquid interface. At the same polymer solution concentration, fluctuations at the interface of the two phases can lead to an unstable flow of the two polymer solutions, and the slip velocity is an important factor affecting the fluctuations between the liquid–liquid interface. After the concentration of the solution is determined, a slow motor speed prevents fiber formation. As the the motor speed gradually increases, the polymer solution flow is gradually stabilized and the collected fiber morphology and quality are significantly improved; when the speed exceeds a certain range, the composite nanofibers break on the collection device. Therefore, it is necessary to control the speed within a certain range to make the polymer solution flow more stable, and the morphology and quality of the prepared composite fibers can improve. Moreover, when the speed is too large, the airflow near the nozzle is enhanced, which affects the process of gas–liquid slip and the fiber is easily broken. Meanwhile, when the velocity is the same, the smaller the nozzle diameter, the more concentrated the polymer flow velocity distribution, and the fiber diameter also decreases with decreasing nozzle diameter. Therefore, repeated controlled experiments are needed to find the optimal parameters for the preparation of composite nanofibers. The morphology, diameter distribution, and thermal decomposition process of the PEO/PVA composite nanofibers are analyzed and compared by SEM experiments and TG experiments. The effects of motor speed and nozzle diameter on the surface morphology and internal structure of the composite nanofibers during fiber preparation are obtained. It is confirmed that the motor speed and nozzle diameter are important factors affecting the polymer slip and the morphology of the composite nanofibers. However, the influence of the slip mechanism of polymer solutions on the structure and performance of composite nanofibers still needs further investigation and refinement. For example, the roughness of the wall surface during the liquid–wall interface slip and the high-speed airflow around the gas–liquid slip affect the preparation and collection of composite nanofibers significantly. Therefore, the selection of suitable experimental parameters is important to ensure composite nanofibers with good morphology for widespread application. Centrifugal spinning for the preparation of composite nanofibers has great potential for improvement. As further research and improvements are made to centrifugal spinning technology, the technology will continue to develop and be more extensively used in various fields.

## Figures and Tables

**Figure 1 nanomaterials-13-01277-f001:**
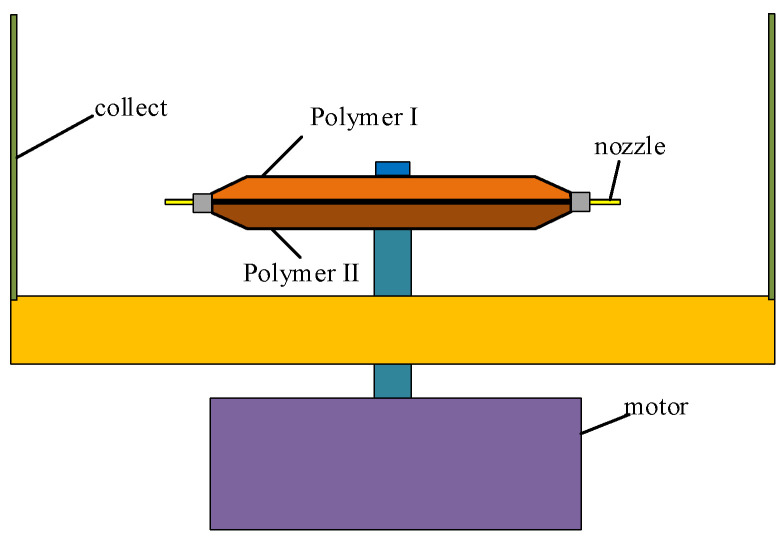
Schematic illustration of the planar structure of high-speed centrifugal spinning device.

**Figure 2 nanomaterials-13-01277-f002:**
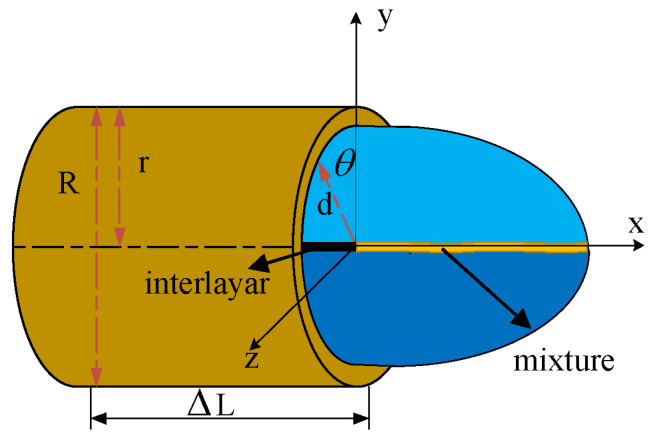
Slip parameter model of nozzle wall.

**Figure 3 nanomaterials-13-01277-f003:**
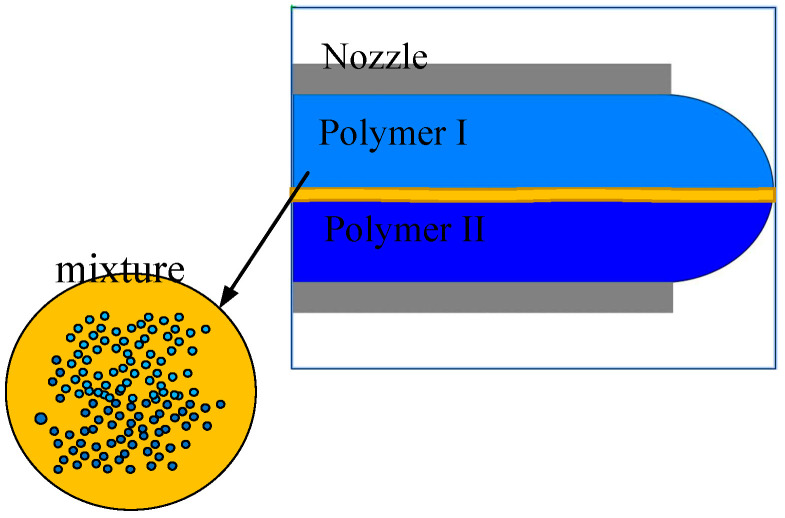
Schematic diagram of two-phase mixed layer.

**Figure 4 nanomaterials-13-01277-f004:**
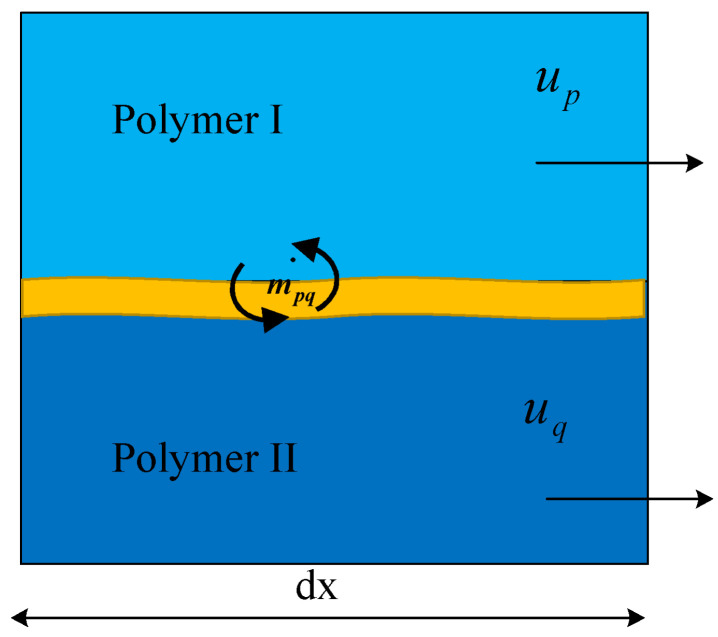
Schematic diagram of two-phase slip.

**Figure 5 nanomaterials-13-01277-f005:**
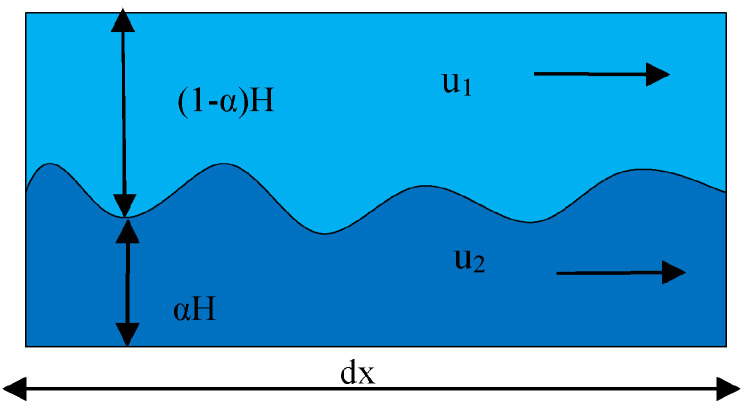
Power wave between liquid–liquid two-phase interface.

**Figure 6 nanomaterials-13-01277-f006:**
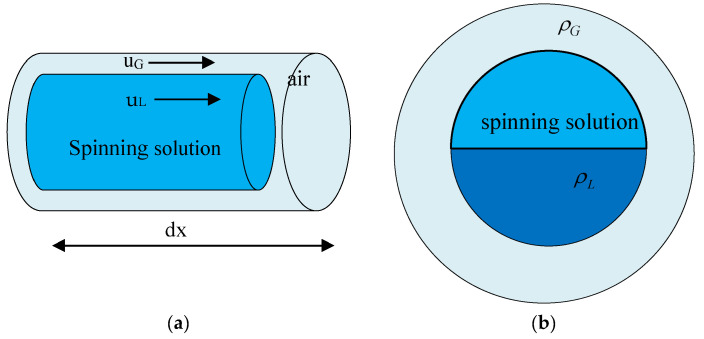
Schematic diagram of polymer gas–liquid slip. (**a**) Schematic diagram of intercepted fiber gas–liquid slip. (**b**) Schematic diagram of gas–liquid interface.

**Figure 7 nanomaterials-13-01277-f007:**
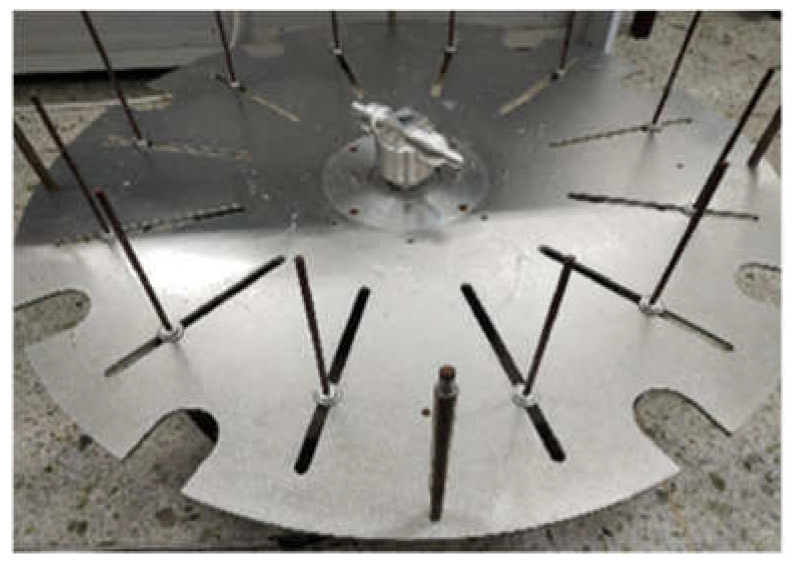
High-speed centrifugal spinning equipment used in the experiments.

**Figure 8 nanomaterials-13-01277-f008:**
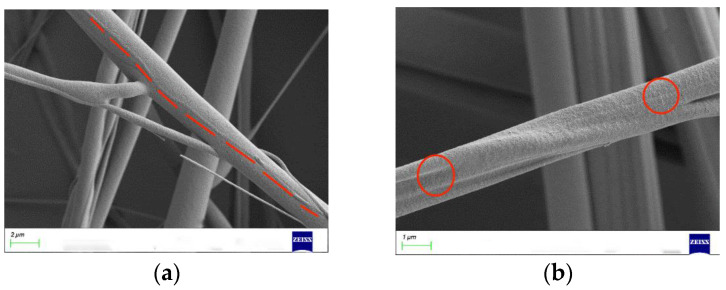
PEO/PVA composite nanofibers morphology with different speeds. (**a**) 1200 rpm; (**b**) 1500 rpm.

**Figure 9 nanomaterials-13-01277-f009:**
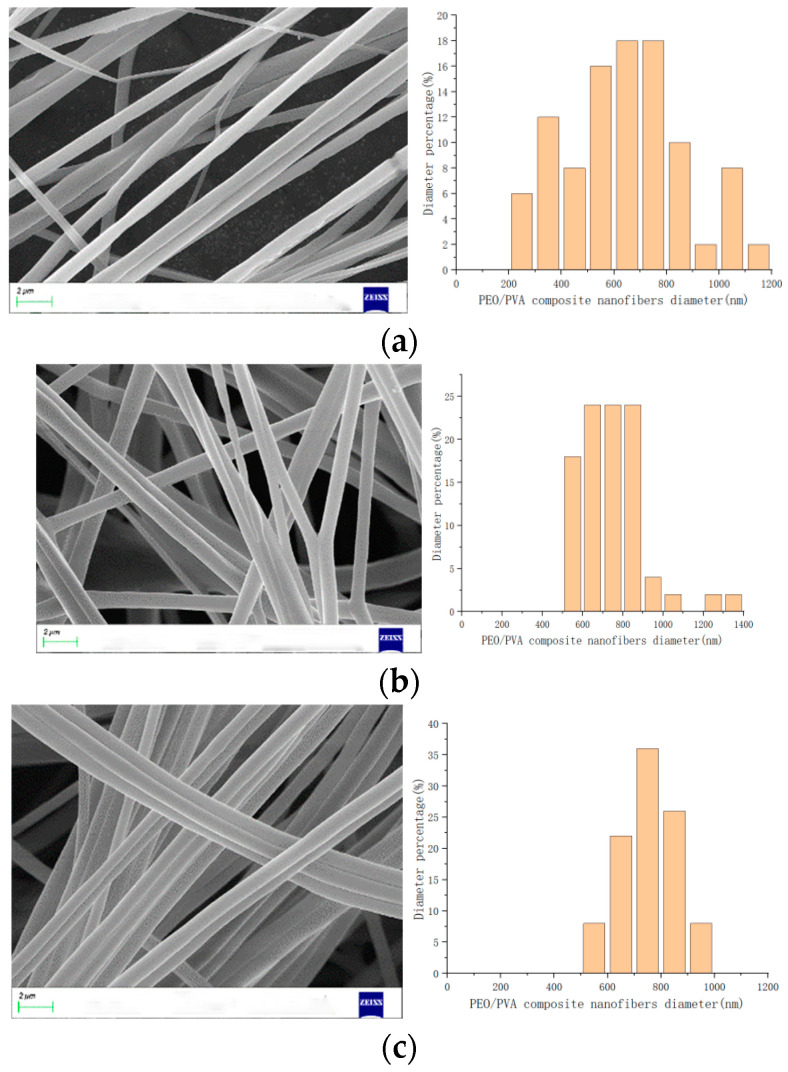
PEO/PVA composite nanofibers with different speeds. (**a**) 1500 rpm; (**b**) 2100 rpm; (**c**) 2700 rpm.

**Figure 10 nanomaterials-13-01277-f010:**
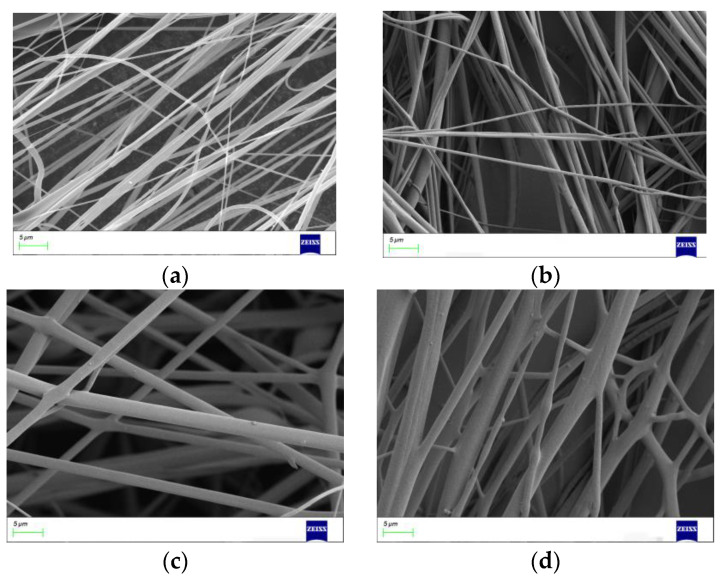
PEO/PVA composite nanofibers at different nozzle diameters. (**a**) 0.2 mm; (**b**) 0.4 mm; (**c**) 0.6 mm; (**d**) 0.8 mm.

**Figure 11 nanomaterials-13-01277-f011:**
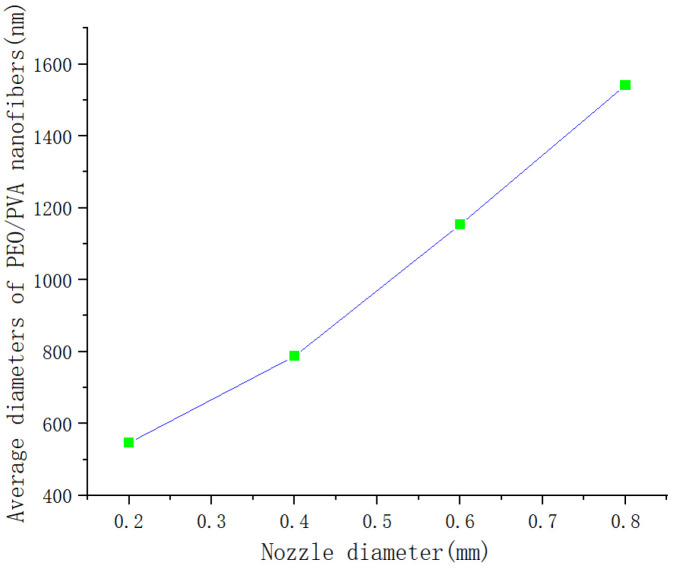
Average diameter of PEO/PVA composite nanofibers versus nozzle diameter.

**Figure 12 nanomaterials-13-01277-f012:**
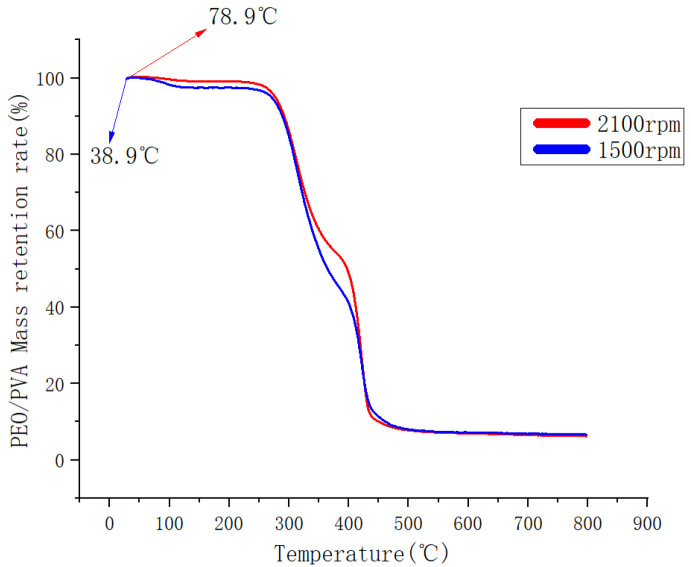
TG curves of PEO/PVA composite nanofibers.

## Data Availability

All individuals included in this section have consented to the acknowledgement.

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
