# Peer review of "High-Speed Centrifugal Spinning Polymer Slip Mechanism and PEO/PVA Composite Fiber Preparation"

_nanomaterials, 2023, doi:10.3390/nano13071277_

Round 1

Reviewer 1 Report

The review section can be improved and works related to the combination of electrospinning and electrospraying can be added 10.1021/acsabm.0c00595

Carbon nanostructures derived from CVD technology and microwave synthesis can be used to improve the properties of polymer fibers doi.org/10.1134/S1070363222060329. Application of carbon nanotubes in polymers improves electrical conductivity https://doi.org/10.3390/jcs6110333

2.It is necessary to formulate the purpose and objectives of the research more specifically.

3.Formulate conclusions according to the objectives and add numerical data.

Add numerical data to the abstract.

4.For section 2, it is necessary to give a summary either in tabular or graphical.

5.The curves shown in Figure 11 have similar values - it is necessary to give an estimate of the error of measurements.

Author Response

Response to Reviewer 1Comments

Point 1: The review section can be improved and works related to the combination of electrospinning and electrospraying can be added 10.1021/acsabm.0c00595. Carbon nanostructures derived from CVD technology and microwave synthesis can be used to improve the properties of polymer fibers doi.org/10.1134/S1070363222060329. Application of carbon nanotubes in polymers improves electrical conductivity https://doi.org/10.3390/jcs6110333.

Response 1: Works related to the combination of electrospinning and electrojet were added to reference.

Point 2: It is necessary to formulate the purpose and objectives of the research more specifically.

Response 2: The purpose and objectives of this study have been improved in the introduction and conclusion.

Point 3: Formulate conclusions according to the objectives and add numerical data. Add numerical data to the abstract.

Response 3: Added numerical data to the summary and conclusion.

Point 4: For section 2, it is necessary to give a summary either in tabular or graphical.

Response 4: The slip model is summarized at the end of Section 2.

Point 5: The curves shown in Figure 11 have similar values - it is necessary to give an estimate of the error of measurements.

Response 5: Because the comparison is the rate of thermal decomposition of the sample, the remaining mass of the sample may be similar at one point in time, so no estimate of the measurement error is given.

Reviewer 2 Report

1. English text should be improved. Introduction and Conclusion have to be rewritten. The aims of the paper should be formulated more clearly in the end of Introduction.

2. Abstract, at the beginning

“The preparation of composite nanofibers by high-speed centrifugal spinning is a novel preparation method.”

Should be replaced by

“The production of composite nanofibers by high-speed centrifugal spinning is a novel preparation method.”

3. Abstract, at the end

“Theoretical and data support was provided for the centrifugal composite spinning slip mechanism and the preparation of composite nanofibers.”

Should be substituted with

“Theoretical evaluation and experimental data were provided for the centrifugal composite spinning slip mechanism and for the preparation of composite nanofibers.”

4. The authors should correct the references according to journal requirements. Please add all authors. Some references are without pages, etc.

5.

Please separate the text in the sections Abstract, Introduction, Experimental, Results and Discussion, Conclusions.

Author Response

Response to Reviewer 2 Comments

Point 1:  English text should be improved. Introduction and Conclusion have to be rewritten. The aims of the paper should be formulated more clearly in the end of Introduction.

Response 1:  Improvements have been made to the English text. Significant revisions have been made to the introduction and conclusion which have been marked in red in the text. The purpose of the paper was revised at the end of the introduction.

Point 2: Abstract, at the beginning “The preparation of composite nanofibers by high-speed centrifugal spinning is a novel preparation method. ”Should be replaced by “The production of composite nanofibers by high-speed centrifugal spinning is a novel preparation method.”

Response 2: Has been replaced as required.

Point 3: Abstract, at the end “Theoretical and data support was provided for the centrifugal composite spinning slip mechanism and the preparation of composite nanofibers. ”Should be substituted with “Theoretical evaluation and experimental data were provided for the centrifugal composite spinning slip mechanism and for the preparation of composite nanofibers.”

Response 3: Has been replaced as required.

Point 4: The authors should correct the references according to journal requirements. Please add all authors. Some references are without pages, etc.

Response 4: Corrections have been made to the references.

Point 5: Please separate the text in the sections Abstract, Introduction, Experimental, Results and Discussion, Conclusions.

Response 5: Corrections have been made as requested.

Reviewer 3 Report

The work is intersting but there are numeorus drawbacks. Fo rexample, any spectroscopis analysis of formulated fibres are reported here.

The work contains large theoretical calculations and reduced number of exterimets. There need to enrich the experimental part, for examples with spectroscopy, DSC, mechnaical test analysis.

The work could be reconsidered after major revision.

Author Response

Point 1: The work is intersting but there are numeorus drawbacks. Fo rexample, any spectroscopis analysis of formulated fibres are reported here. The work contains large theoretical calculations and reduced number of exterimets. There need to enrich the experimental part, for examples with spectroscopy, DSC, mechnaical test analysis. The work could be reconsidered after major revision.

Response 1: In this paper, we focus on the effect of polymer slip mechanism on the preparation of composite nanofibers, which is modeled by extensive theoretical calculations, and finally the theoretical model is verified by analyzing the fiber morphology and quality through SEM and TG experiments. These two experiments can best reflect the comparison of fiber morphology and quality. Due to the softness of the collected fibers and the length of the experiments in the article, no other experiments were used. We will improve this shortcoming and apply these experiments to future studies and experiments.

Round 2

Reviewer 3 Report

The quality of paper has been imporved, but there are not additional experimental data.